# Psychological consultation apps in Saudi Arabia: A study for experts' evaluation and users' points of view

**Abdulmohsen Saud Albesher**⬤*, **Maymunah Faheem Alwahib**

Department of Information Systems, College of Computer Sciences and Information Technology, King Faisal University, Hofuf, Saudi Arabia

* aalbesher@kfu.edu.sa

## Abstract

Medical consultation applications (apps) have rapidly proliferated globally. One type of app is the psychological consultation app, which has made visiting doctors more convenient, particularly for individuals who feel embarrassed about consulting a psychiatrist. However, only a few researchers have examined the usability or user experience of such apps. This study aims to evaluate the user experience of psychological consultation apps in Saudi Arabia, specifically focusing on the usability aspects and user satisfaction of the apps "Labayh," "Estenarah," and "Mind." The research employs two methodologies: First, an expert evaluation using the SMART heuristic framework, developed to assess the usability of mobile apps by identifying usability issues based on established principles. Results from this method revealed that all three apps faced challenges, particularly in SMART 5 (Each interface should focus on one task) and SMART 10 (Cater for diverse mobile environments). Second, a sentiment analysis of user reviews from app stores was conducted, categorizing feedback into positive and negative reviews. User reviews were collected using Heedzy, an online tool designed for extracting reviews from mobile apps. Data cleaning was performed using Python libraries, which handled missing data and removed duplicate entries. Out of 459 reviews analyzed, 51% were negative, focusing primarily on general dissatisfaction and functionality issues, while 49% were positive, highlighting user appreciation and the innovative concept of online consultations. Specific findings indicated that the "Mind" app had significant usability concerns, receiving a severity rating of 70, with notable issues in error prevention and interface clarity. Recommendations for improvement include enhancing task-focused design, increasing adaptability for diverse mobile environments, and addressing user feedback to refine app functionalities. This research contributes valuable insights for designers aiming to improve the usability of psychological consultation apps in the region.

**Data availability statement:** All relevant data are within the manuscript. The collection and analysis method complied with the terms and conditions for the source of the data.

**Funding:** The authors extend their appreciation to the Deanship of Scientific Research, Vice Presidency for Graduate Studies and Scientific Research, King Faisal University, Saudi Arabia, under Grant KFU252578. It is important to note that the funders had no role in the study design, data collection and analysis, decision to publish, or preparation of the manuscript.

**Competing interests:** The authors have declared that no competing interests exist.

## Section 1. Introduction

In the past decade, the expansion of the mobile age has exceeded expectations, impacting many aspects of human life, including learning, shopping, and healthcare. This development has opened up numerous methods of service delivery. In the healthcare sector, doctors can reach patients anywhere and at any time. Additionally, mobile health ensures continuous healthcare delivery during pandemics, such as the recent COVID-19 crisis, when patients could not physically visit hospitals [1,2].

The most significant healthcare challenges facing modern society are mental health issues [3]. Worldwide, approximately one billion people suffer from mental illnesses, which can range from psychotic and personality disorders to more prevalent conditions such as anxiety and depression [4]. Mental diseases significantly reduce the quality of life and the economic and social contributions of affected individuals. The estimated annual "cost" of the combined detrimental effects on the global economy is USD 2.5 trillion, projected to reach USD six trillion by 2030 [4]. The World Health Organization's Special Initiative for Mental Health (2019–2023) aims to provide access to high-quality and affordable care for mental health disorders [5].

According to the Saudi National Mental Health Survey (SNMHS), 34.2% of Saudis have received a mental health diagnosis at some point in their lives [6]. Anxiety disorders, mood disorders, eating disorders, disruptive behavior disorders, and drug use disorders are the most prevalent mental health conditions in the Kingdom of Saudi Arabia (KSA) [7]. Stigma, ignorance, and a lack of high-quality care are some of the obstacles that prevent individuals from receiving timely and adequate treatment for mental health issues [8–10].

The Saudi government has been intensively working on the digital transformation of the healthcare sector, which is a crucial component of Vision 2030. According to the Vision website, a key aspect of this program is telemedicine consultation. A primary focus of this reform is to improve the provision and outcomes of mental health care, as mental health disorders are increasingly recognized as a significant public health issue in Saudi Arabia [11]. This focus on digital health underscores the necessity for usability studies of mental health apps, as enhancing user experience and accessibility is crucial to ensure that these innovative solutions effectively meet the needs of the population. By evaluating the usability of mental health apps, this research aims to align with the government's vision, ultimately contributing to improved mental health support in the digital age.

In recent years, digital mental health solutions have gained significant attention for their potential to increase accessibility and reduce stigma. This is particularly relevant in countries like Saudi Arabia, where mental health disorders are common, and traditional healthcare delivery faces both logistical and social challenges. Psychological consultation apps offer a promising avenue for connecting patients with licensed professionals remotely, thereby improving access to care in culturally sensitive ways.

Existing literature on telemedicine and psychological health apps has primarily focused on general usability and user satisfaction. For instance, Aldekhyyel et al. [12] conducted a heuristic evaluation of telemedicine apps during the COVID-19

pandemic and identified critical interface issues that impact user experience. Poncette et al. [13] highlighted usability problems in remote patient monitoring systems and advocated for user-centered design improvements. Ebnali et al. [14] and Cho et al. [15] examined how usability influences user engagement in mobile health apps, demonstrating that better design can enhance satisfaction and retention. Furthermore, researchers have emphasized that design flaws may pose safety risks [16–19], and that strong usability is associated with increased interaction and sustained use [20,21].

However, despite these contributions, there remains a significant gap regarding the usability and design of psychological consultation apps in the Saudi context. Most existing studies have focused on general telemedicine solutions or specific user groups, such as healthcare workers [22] and mental health patients [23], without addressing the unique cultural and linguistic needs of Saudi users. Aldaweesh et al. [24] explored the broader challenges of digital mental health in Saudi Arabia, while Alqhatani [25] examined privacy policies and their implications for user trust, an important but separate dimension of usability. None of these studies offers a comprehensive, app-level usability evaluation that accounts for both expert perspectives and real-world user feedback.

Therefore, this study aims to address these critical gaps by evaluating the usability of three widely used psychological consultation apps in Saudi Arabia (i.e., Labayh, Mind, and Estenarah). We employ a dual-method approach that integrates heuristic evaluation with sentiment analysis of user reviews. This combination allows us to capture both professional usability assessments and real user experiences, providing culturally grounded insights into app performance. The study advances our understanding of digital mental health tools in the region and delivers practical guidance for designers and health authorities seeking to improve mental healthcare delivery.

The objectives of this study are as follows:

- To evaluate the usability of psychological consultation apps from an expert perspective.

- To identify the issues faced by users of psychological consultation apps.

- To highlight the positive aspects of psychological consultation apps.

- To provide helpful recommendations for the designers of psychological consultation apps.

This paper addresses the following research questions:

1. To what extent are psychological consultation apps in Saudi Arabia usable?

2. What is the user experience of psychological consultation apps in Saudi Arabia?

To answer the research questions, the authors first investigated the available psychological consultation apps in Saudi Arabia. They then filtered them based on their likely suitability for the research. For instance, the authors double-checked whether they offered medical consultations and whether users had written comments on the apps. Finally, the authors identified three apps suitable for the research (Labayh, Estenarah, and Mind). The authors answered the first question by exploring the usability of psychological consultation apps using heuristic evaluation, which studied the experts' perspectives. Then, the authors answered the second question by analyzing the positive and negative user reviews of the apps available in the App Store and Google Play for the years 2022–2023.

This research evaluates psychological consultation apps in Saudi Arabia and provides significant contributions to the fields of mobile health and user experience design. By evaluating three prominent apps through expert heuristic analysis and user review sentiment analysis, the study offers a comprehensive assessment of usability from both expert and user perspectives. It identifies critical usability issues and categorizes user feedback into positive and negative dimensions, revealing insights into user satisfaction and areas needing improvement. This research not only addresses the gap in the literature regarding psychological apps in Saudi Arabia but also supplies actionable recommendations for developers to enhance the design and functionality of these apps. Ultimately, the findings contribute to the ongoing discourse on telemedicine and its role in improving mental health care accessibility, particularly in culturally sensitive contexts.

   

The remainder of the paper is organized as follows: Section 1 provides more information about remote medicine. Section 2 reviews the related literature, with the first subsection addressing telemedicine apps and the second focusing on psychological apps. Section 3 describes the methodology used in both the heuristic evaluation and sentiment analysis. Section 4 presents the results of the two studies. Section 5 provides a detailed discussion, structured into five subsections: a summary of key findings, analysis of usability challenges, identification of positive aspects, evidence-based recommendations, and implications for mental health app design in Saudi Arabia. Section 6 outlines the study's limitations, and Section 7 offers the conclusion.

## Remote medicine

The concept of remote medicine initially emerged in the 1900s [26,27]. Both doctors and patients can save time with online video consultations. Remote medicine is beneficial for patients and healthcare providers who wish to seek a second opinion from experts or who do not have easy access to hospitals [28]. Through smartphone apps, patients can receive support, medical advice, and assistance in emergencies. For instance, patients can use apps to capture images of skin lesions, which can then be sent to a dermatologist who can prescribe suitable medications [29]. Remote medicine has been effectively utilized to prevent, detect, diagnose, monitor, and forecast the course of various diseases without the need for wires, invasive procedures, or in-person contact with medical professionals [30]. Numerous medical and therapeutic applications, including diabetes management [31], asthma monitoring [32,33], chronic disease [34,35], and age-related diseases [36,37], employ telemedicine systems. In critical care, telemedicine is also used [38,39] to address growing patient demand and intensivist shortages [40].

Smartphone technology serves as a disease screening and monitoring tool with minimal additional costs and potentially higher-quality findings. Examples of this technology include app-integrated wearable sensors [41]. Additionally, the smartphone's technological capabilities, widespread use, accessibility, and increasing ownership rates worldwide position it as a desirable tool for patient self-management and ongoing monitoring of symptoms and vital signs [42–44]. Mobile telemental health refers to the use of cell phones and other wireless devices in psychiatric and mental health contexts. Applications of this type include decision assistance systems, health promotion, ecological momentary assessment, and therapy monitoring and adherence [45].

## Section 2. Related work

### Telemedicine apps

The concept of telemedicine is not new. Pavlopoulos et al. [46] created a versatile restorative device that empowers tele-diagnosis, remote support, and teleconsultation of mobile healthcare suppliers by expert specialists. The device transmits vital signs and images of the patient from the emergency area to the medical consultation area using the GSM mobile communication organize. The specialist at the consultation area assesses patient data and issues instructions to the emergency staff on treatment methods until the patient is transported to the hospital. Telemedicine apps have been found to be valuable for both patients and specialists. Krysta et al. [47] reviewed the literature on telemedicine apps for individuals with intellectual disabilities and found that they facilitate communication between patients and physicians, allow for basic behavioral treatments, encourage patient compliance throughout treatment, offer essential medical education, and track patient information for the physician.

Various studies have examined telemedicine from a usability perspective. Layfield et al. [48] systematically explored patient satisfaction with video-based telemedicine services using FaceTime, BlueJeans, and Doximity platforms between March 25, 2020, and April 24, 2020. The method used in this study was a telehealth usability questionnaire (TUQ). The results revealed that most participants reported high satisfaction with telemedicine. Aldekhyyel et al. [12] evaluated the usability of the User Interface (UI) design for several telemedicine apps in Saudi Arabia that were used during the

COVID-19 pandemic. The apps selected for this study were Seha, Cura, and Dr. Sulaiman Alhabib's app. The authors used Nielsen's 10 usability heuristics and provided specific design recommendations. Fifty-four UI usability issues were identified: 18 issues in "Seha," 14 issues in "Cura," and 22 issues in "Dr. Sulaiman Alhabib." The most violated heuristic principles were user control and freedom, recognition rather than recall, consistency, adherence to standards, and error prevention. Booday and Albesher [49] measured several Saudi apps (Sehha, Sehhaty, Mwid, Tabaud, and Tawakklna) that were used during the COVID-19 pandemic. They employed SMART heuristics specifically designed for assessing mobile apps. Alghareeb et al. [50] evaluated the same apps by analyzing their reviews in app stores, focusing on effectiveness, efficiency, and user satisfaction.

Johnson et al. [51] measured patient satisfaction and telemedicine usability in breast cancer care using a modified tele-health usability questionnaire. The sample for this study included patients in Minnesota from June 15, 2020, to September 4, 2020. The findings indicated that the median patient satisfaction score was 5.5 (IQR: 4.25–6.25), and the median tele-medicine usability score was 5.6 (IQR: 4.4–6.2). Poncette et al. [13] evaluated the usability of a remote patient monitoring system in an intensive care unit (ICU) context, identifying UI problems. The evaluation was conducted using a think-aloud protocol with 10 usability tests involving ICU staff. Data were analyzed from testing sessions using a deductive analysis approach with MAXQDA software. The findings revealed 37 individual usability problems specific to monitoring UI, which could be assigned to six subcodes: usefulness of the system, response time, responsiveness, meaning of labels, function of UI elements, and navigation. Changes in graphics and design were proposed to improve navigation, information retrieval, and spatial orientation. The UI was revised by creating a prototype with a more responsive design and modifications in labeling and UI elements. Lee et al. [52] studied the usability of telemedicine in pediatric gastroenterology, assessing the quality of UI and telemedicine interaction across five domains: usefulness, reliability, ease of use, effectiveness, and satisfaction using the telehealth usability questionnaire (TUQ). Data were analyzed using R statistics. The study concluded that most participants (82%) had no prior experience with telemedicine. The average usability score (on a scale of 1–5) was 3.87 ($\sigma = 0.67$), with the highest domain being the usefulness of telemedicine ($\mu = 4.29$, $\sigma = 0.69$) and physician satisfaction ($\mu = 4.13$, $\sigma = 0.79$), while the lowest domain was reliability ($\mu = 3.02$, $\sigma = 0.87$).

Ebnali et al. [14] studied how mHealth, which exhibits better usability than apps with lower usability, can enhance physical activity participation in breast cancer patients. They employed heuristic evaluation. which consisted of Nielsen's ten heuristics, Shneiderman's eight golden rules of design, and Norman's seven principles. One finding of this study is that the user usability score and the heuristic usability score had a positive correlation. Another finding indicated that patients who rated the app higher in usability demonstrated greater commitment to physical activity. Cho et al. [15] assessed the usability of the MyPEEPS app from both expert and end-user perspectives. They conducted a heuristic evaluation with five informatics experts to identify violations of usability principles and performed user usability testing with real users to identify potential obstacles to their use of the app. The findings showed that the mean scores of the overall severity of the identified heuristic violations rated by experts ranged from 0.4 to 2.6 (0 = no usability problem to 4 = usability catastrophe). Overall, end users completed the associated tasks and provided recommendations to improve the usability of the MyPEEPS Mobile app.

## Psychological apps

Some researchers have studied the usability of psychological apps. Alqahtani and Orji [53] aimed to analyze user reviews of 106 available psychological health apps to reveal their strengths, weaknesses, and gaps, thereby identifying why users are likely to discontinue using these applications. A secondary objective was to use insights from the reviews to offer recommendations for designing effective psychological health apps that are more usable and engaging. The methodology employed was a thematic analysis of the reviews. The analysis revealed that users focused more on the UI and user-friendliness of the app, appreciating those that provided a variety of options, functionalities, and content to choose from.

Conversely, poor usability emerged as the most common reason for abandoning psychological health apps. Wong et al. [54] identified factors affecting postsecondary students' attitudes and behaviors when using the Thought Spot psychological health app, describing how these factors influence user experience and subjective user engagement. During a randomized trial, users participated in one-on-one semi-structured interviews that explored their overall experiences and perceptions of the app, as well as factors affecting their usage. The results indicated that users' engagement with the app was influenced by several factors, one of which was the user experience of the app.

Stawarz et al. [55] analyzed the functionality and user opinions of 31 mobile apps that support cognitive behavioral therapy for depression. They identified key factors affecting user experience and supporting engagement. The methodology involved a thematic analysis of available user reviews of cognitive behavioral therapy apps for depression. The results from 1,287 reviews indicated that cognitive behavioral therapy apps for depression require improvement. Furthermore, user experience is influenced by factors such as the context in which the app is used.

Oyebode et al. [56] evaluated 104 psychological health apps on Google Play and the App Store by performing a sentiment analysis of 88,125 user reviews using machine learning (ML). They conducted a thematic analysis of positive and negative reviews to identify themes representing various factors affecting the effectiveness of psychological health apps, both positively and negatively. The results revealed 21 negative themes and 29 positive themes. The negative themes included usability concerns, content concerns, ethical concerns, and billing concerns. Among the positive themes were aesthetically pleasing interfaces and app customization. Finally, design recommendations were provided on how to address the identified negative factors to improve the effectiveness of psychological health apps.

## Section 3. Methodology

This study employs a mixed-methods approach, integrating qualitative and quantitative research elements to assess the usability and user experience of psychological consultation apps. The first study utilizes heuristic evaluation, a qualitative method involving expert reviews to identify usability issues, while the second study applies sentiment analysis, a quantitative technique that analyzes user reviews to quantify sentiments and categorize feedback. The following subsections detail these methodologies: Section 3.1 presents the heuristic evaluation process, and Section 3.2 outlines the sentiment analysis of user reviews, including data extraction, cleaning, and the categorization of sentiments into distinct themes.

### Study one: Heuristic evaluation (expert review)

The methodology of the first study was heuristic evaluation, a popular method for identifying major usability problems in new apps [57–59]. Researchers have indicated that heuristic evaluation is a highly effective, inexpensive, and rapid usability method. However, it involves a degree of subjectivity, as findings rely on the expertise and judgment of the evaluators. Additionally, this method does not involve real user interactions, which may provide more nuanced insights into actual user behaviors and experiences [60,61]. Moreover, heuristic evaluation has been employed in similar research [62–67]. This study applied SMART heuristics because they are specifically designed for smartphones [49]. SMART heuristics are shown in Table 1. Five experts were contacted for the evaluation of the tested apps. Nielsen indicated that 3–5 experts can identify approximately 75% of usability issues. The decision to include exactly five experts was based on selecting the upper limit of this optimal range, as involving more than five experts typically leads to repetitive findings without adding significant value [68,69]. The experts were carefully selected based on their strong background in the field of computer science, with all holding a Master's degree or higher qualification. They were colleagues and professors with experience in usability evaluation techniques. Experts agreed to participate verbally and in writing via WhatsApp. This agreement was obtained after explaining the procedures of the methodology, which they were already familiar with as common in Human-Computer Interaction. The experts received an evaluation form containing the following information: the SMART heuristics, a description of each heuristic, a rating section with five levels (0 = not a usability problem at all, 1 = superficial problem, 2 = minor problem, 3 = major problem, 4 = usability disaster), and a comments section. After the

**Table 1. SMART heuristics [56].**

| SMART# | Heuristic |
|--------|-----------|
| 1 | Provide immediate notification of application status. |
| 2 | Use a theme and consistent terms as well as conventions and standards familiar to the user. |
| 3 | Prevent problems where possible; assist users when a problem occurs. |
| 4 | Display an overlay pointing out the main features when appropriate or requested. |
| 5 | Each interface should focus on one task. |
| 6 | Design a visually pleasing interface. |
| 7 | Intuitive interfaces make for easier user journeys. |
| 8 | Design a clear navigable path to task completion. |
| 9 | Allow configuration options and shortcuts. |
| 10 | Cater for diverse mobile environments. |
| 11 | Facilitate easier input. |
| 12 | Use the camera, microphone, and sensors when appropriate to lessen the user's workload. |
| 13 | Create an aesthetic and identifiable icon. |

experts completed their evaluations, the researchers conducted individual meetings with them when clarification or further discussion about the identified usability issues was necessary. These meetings were held independently with each expert to address specific points and ensure clarity. In the end, the data were analyzed based on the rating problems and the comments provided by the evaluators. The total severity rating for each app was calculated by summing all the resulting values from all the heuristics related to the app. The average for each app was calculated by dividing the severity rating of the app by the number of evaluators, which was five.

## Study two: User reviews

The study aimed to explore users' perceptions of psychological mobile apps in Saudi Arabia by analyzing user reviews. Our methodology consists of five steps, as shown in Fig 1. In the first step, we identified three available Saudi psychological apps from the iOS platform: Labayh, Estenarah, and Mind. In the second step, all available data from August 2017 to September 2023 were extracted from the App Store using Heedzy, an online tool specifically designed for extracting mobile app reviews [70]. A total of 448 reviews were collected from the selected three apps: Labayh (386 reviews), Estenarah (50 reviews), and Mind (12 reviews). The collection and analysis method complied with the terms and conditions for the source of the data. In the third step, the researchers applied several preprocessing techniques using the pandas library, a Python library used for data analysis, including cleaning, filtering, sorting, and visualizing data [71]. The cleaning process began with the removal of redundant data, followed by addressing missing values. Absent titles were replaced with a placeholder ("No Title") to retain all records for analysis. Following that, we excluded all irrelevant reviews (i.e., reviews not related to usability). Then, we applied tokenization by combining the title and content columns into a single unified review for each entry. Next, we performed vectorization of tokenized reviews to convert the text data into a numerical format for machine learning requirements. The preprocessing concluded with the application of the Count Vectorizer method to represent documents based on word counts. Count Vectorizer is considered a model and a feature extraction technique in Natural Language Processing (NLP), used to convert text into numerical data by counting word occurrences [72]. In the fourth step, the data was categorized based on sentiment using user ratings. Reviews with a rating of 1 or 2 out of 5 were indicative of negative sentiment, while the others were considered positive. In the last step of

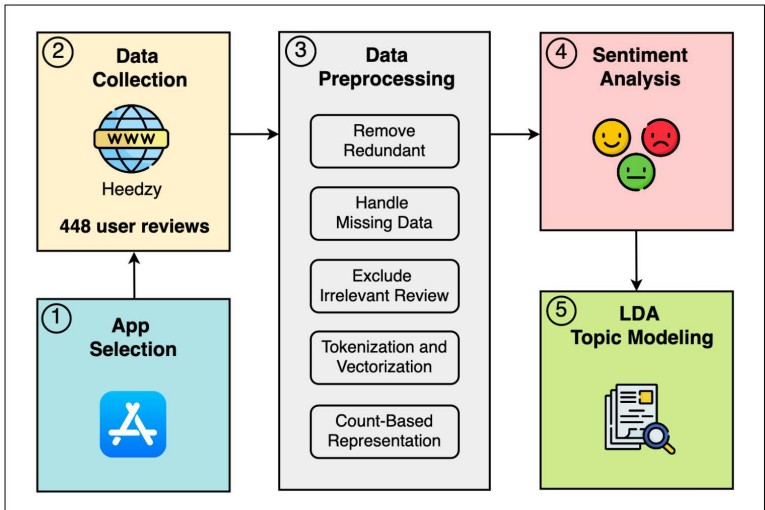

**Fig 1. The methodology steps for the second study.**

our methodology, the Latent Dirichlet Allocation (LDA) method was used to identify potential topics based on the distribution of words in the documents. The LDA model was trained on the processed data using a Count Vectorizer, with key parameters such as alpha and beta adjusted to optimize the model's performance. The implementation of LDA involved several steps. First, we constructed a document-term matrix where each document (review) was represented by its word frequency. Second, we set the hyperparameters for LDA, including the number of topics (alpha), which controls document-topic distribution, and $\beta$ (beta), which controls word-topic distribution. Third, we trained the LDA model to iteratively refine the topic distributions. Fourth, we extracted and analyzed the most representative words for each topic by sorting words based on their probability of belonging to a given topic. Finally, to provide interpretability, the top words for each topic were identified to understand emerging themes in user reviews [73]. To provide more clarity, the top 15 words for each topic were listed, giving a better overview of each topic's scope. The reliability of the sentiment classification was ensured by validating the results with accuracy measures and using a validation set.

While the sub-themes identified from the LDA topic modeling are qualitatively meaningful, the overall sentiment analysis process used in this study is quantitative. Specifically, after the Count Vectorizer converted the text data into a numerical document-term matrix, the LDA model applied probabilistic topic modeling to generate topic distributions across the corpus. Each review was assigned to its most probable topic based on its posterior probability. The percentages shown in Figs 2 and 3 represent the proportion of total reviews associated with each of the dominant topics. Thus, the categorization and distribution calculations are derived from quantitative outputs produced by the trained LDA model.

## Section 4. Results

### Heuristic evaluation

Table 2 demonstrates the results of the first study. Generally speaking, the three apps did not exhibit major issues or significant flaws. It can be observed that the Labayh app had the lowest severity rating, with a total of 44. In contrast, Mind had the highest severity rating (70), indicating significant usability issues. In the context of this study, higher severity ratings for each heuristic indicate more significant usability issues identified by the evaluators, while lower ratings reflect fewer or less severe usability problems. The Mind app exhibited the most severe usability issues, as reflected in high ratings for SMART 1 ("Providing notifications") and SMART 3 ("Error prevention"). For SMART 1, evaluators noted

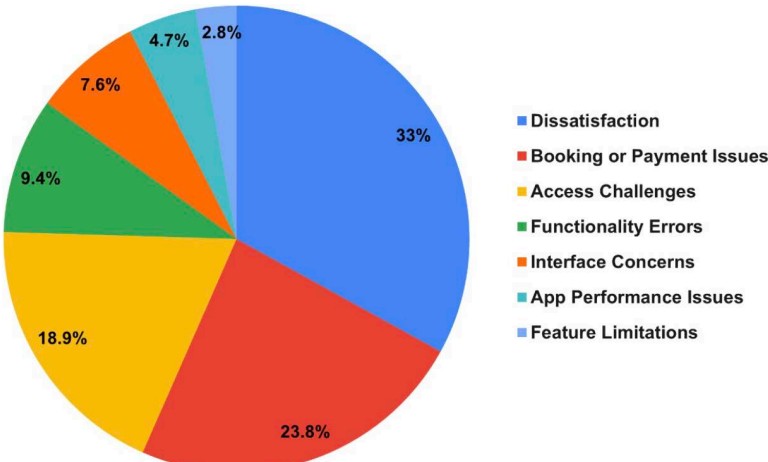

**Fig 2. Negative reviews categories.**

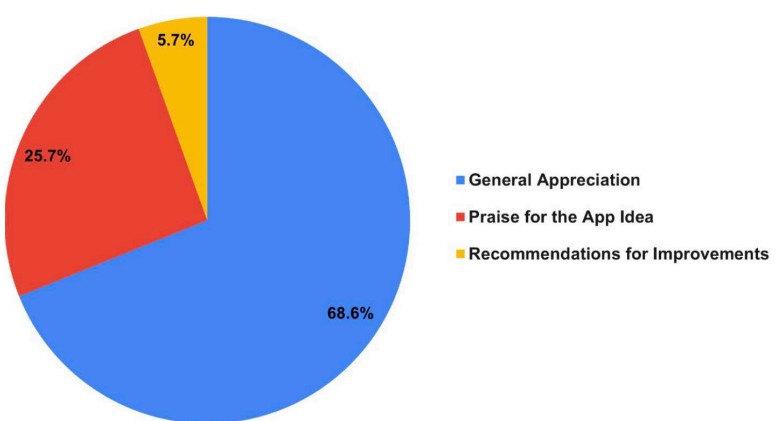

**Fig 3. Positive reviews categories.**

inconsistent and delayed notifications, which led users to miss critical updates, such as reminders for consultations. Regarding SMART 3, the app lacked clear error messages and guidance, such as when users entered invalid payment details, creating frustration as users were unsure how to resolve the issue. Labayh had fewer usability problems overall but faced challenges with SMART 9 ("Configuration options and shortcuts"). The app lacked sufficient customization features, such as the ability to adjust text size or save frequently used inputs, limiting its adaptability to diverse user needs. Estenarah's usability issues were most evident in SMART 5 ("Focus on one task per interface"). The app frequently combined multiple functions, such as booking and payment, on the same screen, increasing cognitive load and leading to user confusion when navigating the app. Estenarah's usability issues were most evident in SMART 5 ("Focus on one task per interface"). The app frequently combined multiple functions, such as booking and payment, on the same screen. This increased cognitive load and led to user confusion when navigating the app.

**Table 2. The results of combining the five experts' evaluations.**

| Heuristic | Labayh | Estenarah | Mind |
|---|---|---|---|
| SMART1 | 1 | 3 | 5 |
| SMART2 | 1 | 5 | 7 |
| SMART3 | 1 | 1 | 9 |
| SMART4 | 4 | 5 | 7 |
| SMART5 | 8 | 8 | 7 |
| SMART6 | 1 | 3 | 5 |
| SMART7 | 1 | 2 | 5 |
| SMART8 | 3 | 5 | 8 |
| SMART9 | 8 | 4 | 5 |
| SMART10 | 9 | 7 | 6 |
| SMART11 | 1 | 1 | 2 |
| SMART12 | 4 | 4 | 2 |
| SMART13 | 2 | 3 | 2 |
| **Total** | **44** | **51** | **70** |
| **Average** | **8.8** | **10.5** | **14** |

## Users' reviews

Reviews were classified as positive or negative. Fig 2 shows the categories of negative reviews. The categories of negative reviews include general dissatisfaction, booking and payment issues, access issues, functionality errors, interface concerns, app performance, and feature limitations. The first category (general dissatisfaction) encompasses feedback about the app's overall quality and functionality. For example, one user wrote, "The app is bad," indicating a general negative sentiment without referencing specific features. The second category (booking and payment issues) includes feedback about difficulties with booking appointments or making payments. For instance, one user wrote, "I tried to book an appointment, but the app charged me without confirming the booking," indicating frustration with the app's payment and scheduling functionality. The third category (access issues) highlights problems that users faced when accessing the app or its features. For example, one user noted, "I couldn't log in after resetting my password; it kept showing an error," demonstrating technical challenges that prevented users from effectively using the app. The fourth category (functionality errors) covers feedback about app features not working as expected. For instance, a user reported, "The video call kept dropping, and I had to restart the app multiple times," indicating issues with the app's core functionality, which affects the user experience. The fifth category (interface concerns) includes feedback about the app's user interface being difficult to navigate or understand. For example, one user commented, "The font on the consultant's profile page is too small to read," reflecting usability concerns related to design and accessibility. The sixth category (app performance) focuses on delays, crashes, or other performance-related problems. For instance, one user stated, "The app crashes every time I try to load my consultation history," highlighting reliability issues that disrupt the user experience. The last category (feature limitations) involves feedback about missing or insufficient app features. For example, a user remarked, "Apple Pay is not supported, which makes payments inconvenient," indicating dissatisfaction due to the app's lack of expected functionalities.

Fig 3 shows the positive reviews. Positive reviews were categorized into three categories: general appreciation, praise for the app idea, and recommendations for improvement. The first category (general appreciation) reveals a strong sense of admiration for the application and extremely positive sentiment about the app's quality and features. For example: "Great app" and "Thank you for the amazing app." The second category (praise for the app idea) includes users' impressions of the app's concept, which allows them to conduct online consultations from anywhere. This category was evident

in comments like "A wonderful app that saves your time from waiting in hospitals." The third category (recommendations for improvement) highlights the app's positive appeal, along with some suggestions from users indicating their desire for future updates. For instance, "Excellent app, but I wish there was a written chat feature."

## Section 5. Discussion

Section 5 presents a structured discussion of the findings from both the heuristic evaluation and sentiment analysis. To enhance clarity and coherence, the discussion is organized into five focused subsections. Section 5.1 summarizes the key empirical findings to provide an overview of usability outcomes across the three apps. Section 5.2 outlines major usability challenges, categorized by design themes such as error prevention and interface clarity. Section 5.3 highlights positive aspects that contributed to user satisfaction. Section 5.4 offers targeted, evidence-based recommendations drawn directly from the evaluation results. Finally, Section 5.5 discusses the broader implications of these findings for mental health app design in the Saudi Arabian context.

### Summary of the key findings

The results from the two studies provide a comprehensive evaluation of the three psychological consultation apps: Labayh, Estenarah, and Mind. In the heuristic evaluation, Labayh emerged as the strongest app, receiving the lowest severity rating of 44, indicating minimal usability issues. In contrast, the Mind app displayed significant usability concerns, evidenced by a high severity rating of 70, particularly regarding error prevention and notification clarity. Estenarah fell in between, with a severity rating of 51, showing usability challenges primarily related to task focus. Additionally, user reviews revealed a mix of positive and negative sentiments. While users appreciated the convenience of online consultations, they expressed dissatisfaction with booking and payment issues, technical errors, and interface concerns across all apps. These findings highlight critical areas for improvement and underscore the need for enhancing user experience in psychological consultation applications.

### Usability challenges

**Error prevention and messaging.** Mind demonstrated serious weaknesses in preventing errors and guiding users through the interface. Users encountered ambiguous error messages without actionable guidance, such as unclear responses to failed payments or booking failures. This aligns with prior research showing that clearly phrased, contextual error feedback enhances user satisfaction and system transparency [74]. Estenarah also struggled with consistent feedback mechanisms, particularly during user input.

**Interface clarity and task orientation.** Mind's interface lacked clarity, especially in how users access core features. Users frequently reported confusion when navigating the consultation booking processes, often abandoning tasks due to unclear menus. This issue compounded the lack of task-oriented screens, violating the "one task per screen" principle [75]. Labayh, on the other hand, demonstrated stronger task focus, contributing to its lower severity rating. This supports cognitive load research showing that when mobile interfaces are overloaded, user satisfaction drops significantly [76] and aligns with established mobile heuristics [75]. Additionally, Sutcliffe [77] emphasizes that clean, goal-directed design significantly enhances user engagement, an area where Labayh outperformed the others.

**Adaptability and personalization.** None of the apps included adaptive interfaces. Customization features such as font sizing, dark mode, or layout preferences were missing, which reduces usability across devices. As highlighted by literature [78], adaptive design significantly improves user satisfaction, particularly in mobile-first regions like Saudi Arabia.

**Payment and technical functionality.** Many users expressed frustration with limited payment methods. Only standard card payments were available, while options like Apple Pay, one of the most widely used methods in Saudi Arabia [79], were not supported. Additionally, users reported frequent crashes, especially with Mind. These technical limitations negatively influenced trust and usability. Improving these areas can generate long-term value, as noted by Powell et al.

[80]. To address these problems, robust testing approaches (unit, integration, and automated) should be implemented [81,82], and development teams should adopt structured code review practices [83]. Moreover, optimizing backend performance through efficient resource allocation [84] and reliable system testing [85] is critical.

## Positive aspects

Despite the aforementioned challenges, the apps demonstrated notable strengths. Both experts and users appreciated the familiar structure of the interfaces, which shortened the learning curve for first-time users. Visual appeal was consistently highlighted, with high praise for graphic design, layout, and branding [86,87]. This aligns with findings from Lima and Wangenheim [86] and Eytam [87] on how aesthetics influence engagement and simplicity. Furthermore, Labayh's lack of responsive adaptation could be improved through multi-device interface design principles advocated by Voutilainen et al. [88] and Parlakkiliç [89]. The concept of remote consultation itself was praised, with many users noting how these apps saved time and avoided the discomfort of in-person visits. Fan et al. [90] similarly noted that digital consultations reduce hospital burden and improve patient convenience, supporting findings by Murphy et al. [91] and global studies on the increasing popularity of telehealth [92].

## Evidence-based recommendations

Based on the findings from the heuristic evaluations and sentiment analysis of user reviews, the following targeted recommendations are proposed to address specific usability problems observed in the three apps. These suggestions are tailored to the issues uncovered during the study rather than reflecting general usability best practices:

- Clarify error messaging (Mind and Estenarah): Replace vague and technical error messages with user-friendly, context-aware language and visual guidance [93].

- Simplify task flows and screens (Mind and Estenarah): Adopt a one-task-per-screen design and reduce steps in key processes such as onboarding and booking [75,76].

- Introduce adaptive interfaces (all apps, especially Labayh): Support responsive UI features such as adjustable font sizes, color schemes, and layout preferences to enhance accessibility [78,88].

- Enhance payment flexibility (all apps): Incorporate localized payment options, including Apple Pay, Mada, and bank transfers, in line with Saudi consumer behavior [79].

- Improve technical performance and reliability (Mind): Employ comprehensive testing (unit, integration, automation), conduct code reviews [83], and optimize network handling using established backend performance models [81–85].

- Leverage visual consistency and familiar patterns (all apps): Reinforce intuitive navigation and consistent iconography to maintain and enhance positive user feedback regarding familiarity [75,86].

## Implications for mental health app design in Saudi Arabia

This study confirms that usability flaws in mental health apps are both technical and cultural. Users in Saudi Arabia prefer apps that are accessible, easy to use, and reliable, reflecting their language needs, literacy levels, and familiarity with digital technology. Designers must consider culturally relevant interface choices, including simplified Arabic language, localized content metaphors, and support for social norms regarding privacy and payment. These findings support the work of Alqahtani and Orji [94] and underscore the importance of designing digital health tools that align with the cultural context in which they are deployed. They also align with broader calls for online-offline service integration [92], user-centered HCI principles [95], and optimizing interface design through usability error analysis [96].

## Section 6. Limitations of the study

This study has inherent limitations that must be acknowledged. While the sample size of user reviews is substantial, it may not fully represent the diverse user base of psychological consultation apps across Saudi Arabia. Additionally, the focus on three specific apps may limit the generalizability of the findings to other contexts or app types. Future research should consider a broader range of apps and a more diverse user demographic to validate the findings and enhance their applicability. Another limitation is the potential bias in user reviews, which may not accurately reflect the full spectrum of user experiences. Users with extreme experiences, either very positive or very negative, are often more motivated to leave reviews, potentially skewing the results. Employing mixed-method approaches that include qualitative interviews could provide deeper insights into user experiences and motivations.

## Section 7. Conclusion

This study is among the first to evaluate the usability of psychological consultation apps in Saudi Arabia using a dual-method approach that combines expert heuristic evaluation with sentiment analysis of user reviews. This methodological integration offers a culturally grounded understanding of how users interact with digital mental health platforms in a rapidly transforming healthcare environment. The findings revealed recurring usability challenges related to interface clarity, error prevention, adaptability across devices, and technical performance, while also recognizing strengths such as familiar design patterns and visual appeal. These insights contribute to the design of more effective and accessible digital mental health solutions tailored to the Saudi context. A notable limitation of this study is the absence of real-time user testing, which could have further enriched behavioral insights. Future research should expand the app sample, incorporate participatory and longitudinal evaluation methods, and explore how usability improvements impact actual mental health outcomes. As digital consultations become a cornerstone of mental healthcare in Saudi Arabia, improving app usability will be critical to ensuring equitable access and sustained user engagement.

## Author contributions

**Conceptualization:** Abdulmohsen Saud Albesher.

**Data curation:** Maymunah Faheem Alwahib.

**Formal analysis:** Maymunah Faheem Alwahib.

**Investigation:** Abdulmohsen Saud Albesher.

**Methodology:** Abdulmohsen Saud Albesher, Maymunah Faheem Alwahib.

**Project administration:** Abdulmohsen Saud Albesher.

**Resources:** Abdulmohsen Saud Albesher.

**Supervision:** Abdulmohsen Saud Albesher.

**Validation:** Maymunah Faheem Alwahib.

**Visualization:** Maymunah Faheem Alwahib.

**Writing – original draft:** Abdulmohsen Saud Albesher, Maymunah Faheem Alwahib.

**Writing – review & editing:** Abdulmohsen Saud Albesher.

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
