## [Decision Letter · Decision Letter 0]

PONE-D-24-40629Psychological consultation apps in Saudi Arabia: A study for experts’ evaluation and users’ points of viewPLOS ONE

Dear Dr. Albesher,

Thank you for submitting your manuscript to PLOS ONE. After careful consideration, we feel that it has merit but does not fully meet PLOS ONE’s publication criteria as it currently stands. Therefore, we invite you to submit a revised version of the manuscript that addresses the points raised during the review process.

We look forward to receiving your revised manuscript.

Kind regards,

Najmul Hasan, PhD

Academic Editor

PLOS ONE

Journal Requirements:

2. In your Methods section, please include additional information about your dataset and ensure that you have included a statement specifying whether the collection and analysis method complied with the terms and conditions for the source of the data.

4. Thank you for stating the following financial disclosure: Funding for this work was provided by the Deanship of Scientific Research, Vice Presidency for Graduate Studies and Scientific Research, King Faisal University, Saudi Arabia (Grant No. 3292).

Reviewers' comments:

Reviewer's Responses to Questions

**Comments to the Author**

1. Is the manuscript technically sound, and do the data support the conclusions?

Reviewer #1: Partly

Reviewer #2: Partly

Reviewer #3: Partly

2. Has the statistical analysis been performed appropriately and rigorously? 

Reviewer #1: No

Reviewer #2: N/A

Reviewer #3: I Don't Know

3. Have the authors made all data underlying the findings in their manuscript fully available?

Reviewer #1: No

Reviewer #2: No

Reviewer #3: Yes

4. Is the manuscript presented in an intelligible fashion and written in standard English?

Reviewer #1: Yes

Reviewer #2: No

Reviewer #3: Yes

5. Review Comments to the Author

Reviewer #1: 1. The introduction is too broad and lacks a clear direction toward the research problem. The research gap is not explicitly described, making it difficult to understand the study's contribution.

2. Figures 1 and 2 are mentioned but not included in the paper, which disrupts the flow and leaves readers without critical visual information.

3. The process of generating the results is insufficiently detailed, leaving gaps that confuse readers and hinder their understanding of the methodology.

4. Subsection 4.2 does not adequately represent sentiment analysis. It appears to focus on qualitative categorization rather than the expected complex sentiment analysis process, which needs clarification.

5. The transition to the discussion section is abrupt and confusing. The link between the previous sections and the discussion is weak, and the structure requires significant improvement for better coherence.

6. While the recommendations are insightful, they should align more closely with the results to provide a stronger foundation for their validity.

7. The conclusion is overly simplistic and does not sufficiently elaborate on the findings. It should answer the research questions more explicitly and synthesize the results in greater detail.

Reviewer #2: Dear Authors,

Thank you for your submission. The manuscript has been thoroughly reviewed, and several suggestions for improvement have been provided. Please carefully consider the feedback and revise the manuscript accordingly. Make sure to highlight all changes in the text for easier review. We look forward to receiving the revised version and appreciate your efforts to enhance the quality of your work.

Sincerely,

Reviewer #3: Introduction.

Line 25

1. what does m-health stand for?

Methodology.

Does this study fall into the category of qualitative or quantitative research?

Line 196.

"was to identify how users feel about apps."

- elaborate specifically, what do you mean by how users "feel" about apps? feeling is an emotion, are exploring their emotions? perhaps there are better terminologies than the "feeling" of users.

Both figures 1 and 2 are blurry and unclear. I could not read the texts, even when downloaded. Please provide a better quality of the figures.

Results

Please provide the quotations that support the themes. Every theme must be backed with the quotation of the reviews.

Please provide the demographic profile of reviewers if possible.

6. PLOS authors have the option to publish the peer review history of their article (what does this mean? ). If published, this will include your full peer review and any attached files.

**Do you want your identity to be public for this peer review?** For information about this choice, including consent withdrawal, please see our Privacy Policy .

Reviewer #1: No

Reviewer #2: No

Reviewer #3: No

---

## [Author Response · Author response to Decision Letter 1]

10 Feb 2025

Reviewer #1:

Comment 1: The introduction is too broad and lacks a clear direction toward the research problem.

Response: Thanks for this comment. We moved two paragraphs to a new section (1.1) and removed two broad paragraphs. We also advanced the discussion of the research objectives to an earlier point in the text.

Comment 2: The research gap is not explicitly described, making it difficult to understand the study's contribution.

Response: Thanks for this comment. We completely wrote the part that describes the research gap again. We removed broad references and added references belonging to mental health and Saudi Arabia. We discussed the research gap in a way that makes it specific and clear. We also added one paragraph about the research contribution.

Comment 3: Figures 1 and 2 are mentioned but not included in the paper, which disrupts the flow and leaves readers without critical visual information.

Response: They were included. The other reviewer saw them and commented on their clarity. We redesigned Figure 1 and included more information. We redesigned Figure 2 and divided it into Figure 2 (negative reviews) and Figure 3 (positive reviews). We added a percentage for every review category.

Comment 4: The process of generating the results is insufficiently detailed, leaving gaps that confuse readers and hinder their understanding of the methodology.

Response: Thank you for your comment. We have rewritten the methodology section to include more detailed explanations and have enhanced Figure 1 for improved visual clarity. We believe these revisions will address the gaps you identified and enhance the overall understanding of our methodology.

Comment 5: Subsection 4.2 does not adequately represent sentiment analysis. It appears to focus on qualitative categorization rather than the expected complex sentiment analysis process, which needs clarification.

Response: Thank you for your valuable feedback. We have revised this section to provide a clearer representation of the sentiment analysis process. While we acknowledge the qualitative categorization of reviews, we have emphasized the underlying complex methodologies used in our analysis. The revisions now include a more detailed explanation of our preprocessing techniques, including tokenization and vectorization, which convert text data into a numerical format suitable for analysis. Additionally, we have highlighted the use of the Latent Dirichlet Allocation (LDA) model to identify topics based on word distributions, which adds depth to our sentiment analysis approach. We believe these clarifications will provide a more comprehensive understanding of the sentiment analysis conducted and address the complexities involved.

Comment 6: The transition to the discussion section is abrupt and confusing. The link between the previous sections and the discussion is weak, and the structure requires significant improvement for better coherence.

Response: Thank you for your valuable feedback. We have added a new introductory paragraph for the discussion section that summarizes the findings from both studies. This paragraph highlights the usability evaluations of the apps and incorporates user feedback, creating a comprehensive overview. Additionally, we have made improvements to the methodology and results sections to ensure clarity and coherence. These enhancements facilitate a smoother and more logical transition into the discussion, allowing readers to better understand the implications of the findings. We believe these changes significantly strengthen the overall flow of the paper.

Comment 7: While the recommendations are insightful, they should align more closely with the results to provide a stronger foundation for their validity.

Response: We have revised the recommendations section to ensure a clearer connection between our findings and proposed solutions. Each recommendation now explicitly references specific results from our evaluations, providing a stronger foundation for their validity.

Comment 8: The conclusion is overly simplistic and does not sufficiently elaborate on the findings. It should answer the research questions more explicitly and synthesize the results in greater detail.

Response: Thank you for your insightful comment. We acknowledge that the previous version was overly simplistic and did not delve deeply enough into the findings. In the revised conclusion, we have made a concerted effort to explicitly answer the research questions, synthesizing the results in detail. We highlighted the usability challenges faced by the apps and provided actionable recommendations based on our findings. We incorporated future directions that suggest areas for further research. This enriched conclusion aims to better encapsulate the essence of our research and its implications for the design and functionality of psychological consultation apps in Saudi Arabia.

Reviewer #2:

Abstract

Comment 1: The research goal is not clearly stated, and it is not specified what exact aspects of user experience the study addresses.

Response: Thank you for your feedback. We have clarified the research goal in the revised abstract. The aim is now explicitly stated as evaluating the user experience of psychological consultation apps in Saudi Arabia, focusing on usability aspects and user satisfaction. This should provide a clearer understanding of the study's objectives.

Comment 2: The methodology is not fully explained; the concept of SMART heuristic is vague.

Response: We appreciate your comment. In the improved abstract, we have elaborated on the SMART heuristic framework, explaining its purpose in assessing usability by identifying issues based on established principles.

Comment 3: The results are presented in a general way, without enough detail.

Response: Thank you for this comment. We have enhanced the presentation of results in the revised abstract, providing specific findings, such as the severity rating of the "Mind" app and the key usability issues identified.

Comment 4: Recommendations are not elaborated on, and their impact on improving the apps is not clarified.

Response: We acknowledge your concern. In the updated abstract, we have elaborated on the recommendations for improvement, specifying actions such as enhancing task-focused design and increasing adaptability for diverse mobile environments. This addition aims to clarify their potential impact on improving the apps.

Introduction

Comment 5: The introduction lacks a clear and focused thesis statement. The main objective

of the paper is not explicitly introduced until later

Response: Thanks for this comment. We went through the introduction again and narrowed it by moving two paragraphs to section (1.1) and removing two very broad paragraphs. We also advanced the discussion of the research objectives to an earlier point in the text. These modifications should make reaching the main objective fast and smooth.

Comment 6: The section about the historical development of remote medicine feels disconnected.

Response: Thanks for this comment. We moved this paragraph to section (2.1).

Comment 7: The paragraph discussing the negative impact of COVID-19 on senior citizens' mental health feels somewhat tangential to the core topic of the paper, which is evaluating app usability. While it is relevant to the broader issue of mental health, it does not directly contribute to the discussion about the need for better psychological consultation apps in Saudi Arabia.

Response: Thank you for this comment. Yes, we read it again and find it very broad and does not directly contribute to the discussion. Therefore, we removed this paragraph from the manuscript.

Comment 8: While the introduction mentions the Saudi government’s efforts in digital health

transformation (Vision 2030), the relevance of this to the research is not fully explained. How does the government's focus on digital healthcare connect to the need for usability studies of mental health apps? This could be clarified to strengthen the argument for the significance of the study.

Response: Thanks for this comment. We fixed this issue by strengthening the argument for the connection of this part with the aim of our study. We removed the part that talks about Seha since it lacks a connection to our study goal.

Comment 9: The numerous references to previous studies on telemedicine and psychological apps seem to be a listing of studies without providing a clear comparison or synthesis of their findings. It would be more effective to highlight the gaps in the literature that this study aims to address, particularly the lack of research on psychological consultation apps in Saudi Arabia.

Response: Thanks for this comment. We revised this part. We removed broad references and added focused ones that are related to mental health and Saudi Arabia. We also compared the findings of these studies. We wrote this part again in a way that made the research gap very specific and clear.

Method

Comment 10: The justification for contacting exactly five experts for app evaluation is insufficiently explained.

Response: We added a justification for the selection of five evaluators, referencing Nielsen's guidelines and detailing the evaluators’ qualifications.

Comment 11: More context on experts’ qualifications and selection criteria is needed.

Response: Thank you for your comment. We have added details about the experts' qualifications and the selection criteria.

Comment 12: The heuristic evaluation process explanation is vague; details on evaluator discussions are lacking.

Response: Thank you for your comment. We have clarified the heuristic evaluation process, including details about the individual meetings held with evaluators to discuss and clarify identified usability issues.

Comment 13: Acknowledge the limitations of the heuristic evaluation method regarding subjectivity and real user interaction.

Response: Thank you for your comment. We have acknowledged the limitations of the heuristic evaluation method to provide a balanced view.

Comment 14: Insufficient detail on sentiment analysis methodology; parameters for LDA not described.

Response: Thank you for your comment. We have added details about the sentiment analysis methodology, including the key parameters for LDA and the validation process.

Comment 15: The steps for tokenizing and vectorizing the reviews could be further elaborated, including the handling of stopwords, stemming, or lemmatization.

Response: We have elaborated on the steps for tokenizing and vectorizing the reviews, including the use of Count Vectorizer, LDA parameters, and the process for identifying key topics and validating the sentiment classification.

Comment 16: Data collection criteria for user reviews are unclear; specifics on the filtering process and review selection are needed.

Response: Thank you for your comment. By "limited," we mean that the number of user reviews was small, and all available reviews were included in the analysis without any filtering or categorization.

Comment 17: Details on data cleaning methods using Python libraries not provided.

Response: We have provided additional details on the techniques used, including the library used, handling missing data, removing duplicates, categorizing reviews, and validating the dataset.

Comment 18: Categories for reviews (seven negative, three positive) need clearer explanation and derivation.

Response: Thank you for your feedback. We have updated the methodology section to include a detailed explanation of how the categories were derived.

Comment 19: The binary classification of reviews oversimplifies sentiment; a more nuanced approach is suggested.

Response: Thank you for your feedback. Our approach aligns with established practices in exploratory studies where rating-based classification provides a straightforward method to identify general sentiment trends. While we acknowledge the nuances in some reviews, this method is practical for capturing broad patterns. Future work can incorporate advanced sentiment analysis for deeper insights.

Results

Comment 20: Heuristic evaluation results lack statistical analysis; no significance tests were conducted.

Response: The primary aim of our study was to explore usability issues and trends rather than conduct a statistical comparison. The descriptive analysis provided sufficient insights to identify the apps with the most significant usability concerns and allowed us to offer targeted recommendations for improvement.

Comment 21: Interpretation of severity ratings needs clarity; what "high" ratings mean should be explained.

Response: Thank you for your comment. We provided a detailed interpretation of what "high" ratings mean.

Comment 22: Fragmented narrative on heuristics and their usability issues; needs better organization.

Response: Thank you for your feedback. We have reorganized Section 4.1 by grouping findings by app and providing detailed explanations of specific heuristics.

Comment 23: Vague explanations of negative review categories; specific examples are needed for clarity.

Response: Thank you for your comment. The paper already includes descriptions and examples for each category. However, we have revised the text to make the examples more significant and added further clarification about the classification.

Comment 24: Counts of negative and positive reviews lack context; methodology for selection unclear.

Response: Thank you for your feedback. We have revised the methodology section to include the number of reviews for each app. Additionally, we clarified that we included all available reviews up to September 2023 without applying a specific timeframe.

Comment 25: Redundancy in review categories; hierarchical categorization could improve clarity.

Response: Thank you for your suggestion. We acknowledge the potential for overlap in our categories; however, our approach aimed to maintain specificity by highlighting specific themes such as "access issues" and "interface concerns" separately. This allows us to provide a more detailed and actionable analysis of user feedback. Reorganizing the categories into a hierarchical structure at this stage could affect the consistency of our findings and recommendations. However, we appreciate your insight and will consider exploring this approach in future studies to improve clarity.

Comment 26: Inconsistent terminology used; definitions for categories needed for better understanding.

Response: Thank you for your comment. We have revised the results section to include clearer definitions for each category of user reviews. We also provided specific examples of user feedback.

Comment 27: The presentation of results could benefit from visual representations (charts/graphs) for clarity.

Response: Thank you for your comment. We have updated the representation of the results by creating a pie chart that displays the percentage distribution for each category.

Discussion

Comment 26: The lack of deep analysis and interpretation of the results.

Response: The revised discussion now includes a more thorough analysis of the usability challenges faced by the "Mind" app, detailing specific issues such as unclear navigation and poor error prevention. The explanation of how these factors contribute to user frustration was expanded.

Comment 27: No detailed analysis is provided on what led to these problems or how they can be rectified.

Response: The revised text now includes specific factors contributing to the usability issues in the "Mind" app, such as vague error messages and the need for a task-focused design. Recommendations for improvement were also incorporated.

Comment 28: The discussion does not address the limitations of the study.

Response: A section addressing the limitations of the study was added, acknowledging the sample size and potential biases in user reviews, as well as the limited generalizability of the findings.

Comment 29: The discussion briefly mentions usability problems but lacks exploration of how these issues affect user experience.

Response: The revised discussion now details how specific usability challenges, such as error prevention and navigation clarity, affect overall user experie

---

## [Decision Letter · Decision Letter 1]

PONE-D-24-40629R1Psychological consultation apps in Saudi Arabia: A study for experts’ evaluation and users’ points of viewPLOS ONE

Dear Dr. Albesher,

Thank you for submitting your manuscript to PLOS ONE. After careful consideration, we feel that it has merit but does not fully meet PLOS ONE’s publication criteria as it currently stands. Therefore, we invite you to submit a revised version of the manuscript that addresses the points raised during the review process.

We look forward to receiving your revised manuscript.

Kind regards,

Najmul Hasan, PhD

Academic Editor

PLOS ONE

Reviewers' comments:

Reviewer's Responses to Questions

**Comments to the Author**

1. If the authors have adequately addressed your comments raised in a previous round of review and you feel that this manuscript is now acceptable for publication, you may indicate that here to bypass the “Comments to the Author” section, enter your conflict of interest statement in the “Confidential to Editor” section, and submit your "Accept" recommendation.

Reviewer #1: (No Response)

Reviewer #3: All comments have been addressed

2. Is the manuscript technically sound, and do the data support the conclusions?

Reviewer #1: Partly

Reviewer #3: Partly

3. Has the statistical analysis been performed appropriately and rigorously? 

Reviewer #1: No

Reviewer #3: I Don't Know

4. Have the authors made all data underlying the findings in their manuscript fully available?

Reviewer #1: No

Reviewer #3: No

5. Is the manuscript presented in an intelligible fashion and written in standard English?

Reviewer #1: Yes

Reviewer #3: Yes

6. Review Comments to the Author

Reviewer #1: A. Introduction Section

The introduction presents a relevant and timely topic — the usability of psychological consultation apps in Saudi Arabia — and provides a general overview of the context and challenges involved. However, the research gap is somewhat implied rather than clearly articulated. For the introduction to effectively set up the study, the gap in existing literature should be made more visible and explicit. To strengthen this section, I recommend the following:

1. Tighten the narrative flow by placing a clear, concise statement of the research gap earlier in the introduction — ideally before the research questions and objectives are introduced.

2. Clearly state what is missing in the current literature. For example:

o Have previous studies not focused on mental health apps in the Saudi Arabian context?

o Is there a lack of comparative usability analysis using real user reviews?

o Are cultural and contextual usability factors underexplored in prior research?

3. Follow the gap with a direct explanation of why this study matters:

Emphasize how addressing this gap will contribute new knowledge (e.g., culturally grounded usability insights), practical value (e.g., improved app design for better mental health access), or methodological contribution (e.g., using user reviews for usability assessment).

B. Discussion Section

1) The discussion section would benefit from a clearer structural organization. Currently, it merges results, analysis, and recommendations in a way that can obscure the key insights of the study. I recommend dividing this section into clear thematic sub-sections, such as:

• Summary of Key Findings

• Usability Challenges (with potential sub-themes like error prevention, interface design, etc.)

• Positive Aspects

• Recommendations for Improvement

• Implications for Mental Health App Design in Saudi Arabia

2) While the discussion references relevant literature, there is often a lack of clear integration between the empirical findings and these theoretical insights. I suggest more explicitly linking findings from the current usability evaluations with the literature cited. Example Revision: Instead of stating:

“Research indicates that effective error messages are crucial for user satisfaction...”

It would be stronger to write:

“Our findings indicate that Mind’s ambiguous error messages negatively affected user experience. This aligns with Smith et al. [74], who argue that clearly phrased, contextual error feedback enhances user satisfaction and system transparency.”

3) The comparative analysis between the three apps—Labyh, Mind, and Estenarah—could be further developed. While comparative statements exist, they often remain descriptive rather than analytical. Expand on why one app succeeded or failed relative to the others. For instance:

• What specific design choices led to Labyh’s better navigation scores?

• Did Estenarah’s simplicity compromise advanced features?

• What lessons can be drawn from Mind’s struggles with technical issues?

4) There is noticeable repetition of some themes, such as task overload, error prevention, and interface challenges, across multiple sections. Consolidating overlapping discussions could enhance clarity and allow more space for nuanced interpretation. Suggestion: Merge similar topics under broader thematic labels—for example, combine “error messaging” and “technical terminology” under a unified discussion of Error Prevention and Communication.

5) Some parts of the discussion adopt a descriptive tone, which weakens the analytical rigor of the section. Suggestion: Adopt a more evaluative tone to critically reflect on the findings. For example:

“Although Labyh outperformed the other apps in navigation, its poor adaptability to different devices highlights a critical flaw in addressing Saudi Arabia’s increasingly mobile-first digital environment.”

C. Recommendation Section

The recommendations section provides a series of general usability best practices — such as improving error messaging, implementing responsive design, and supporting diverse payment options — which are certainly relevant to the design of psychological consultation apps. However, as it stands, this section lacks a clear, direct link to the empirical findings presented in the earlier parts of the paper.

While the study includes valuable data on usability issues derived from user reviews and comparative app analysis, these insights are not explicitly reflected in the recommendations. This results in a disconnect between the evidence gathered and the guidance offered. For instance, suggestions such as “conduct user journey mapping” or “implement contextual adaptation” are useful but appear to be generic UX advice rather than recommendations tailored to the actual usability problems uncovered in Mind, Labyh, and Estenarah.

Suggestion for Improvement:

To enhance the value and academic rigor of the recommendations, I strongly encourage the authors to restructure this section to reflect the comparative analysis presented earlier. One effective approach would be to build a matrix that clearly maps:

• Usability problems observed in each app

• Comparative insights (e.g., where one app succeeded and another failed)

• Targeted, app-specific recommendations grounded in these observations

D. Conclusion Section

While the conclusion effectively summarizes the key findings, it tends to be overly descriptive and lacks synthesis. Rather than reiterating the results, the conclusion should offer a deeper interpretation of their broader significance — essentially answering the “so what?” of the study. Additionally, the paragraph addresses multiple themes (usability findings, future directions, methodological limitations, sentiment analysis critiques, and reviewer demographics) within a single block, which dilutes clarity and impact.

Consider separating the conclusion into clearer thematic segments. Start with a concise summary of the main findings and their implications, followed by a discussion of methodological limitations, and conclude with forward-looking insights or practical recommendations.

The unique contribution of the study — such as the integration of heuristic evaluation and sentiment analysis within a culturally specific context (Saudi Arabia) — should be highlighted more explicitly.

Reviewer #3: Are you sure the sentiment analysis falls under the category of quantitative approach? The categorization to into several sub-themes such as general dissatisfaction, booking and payment issues, access issues, functionality errors, interface concerns, app performance, and feature limitations (for negative reviews) and general appreciation, praise for the app idea, and recommendations for improvement (for positive reviews) based on the subjective reviews strongly indicates qualitative analysis. How are the percentages of each category as shown in Fig 2 and 3 obtained?

7. PLOS authors have the option to publish the peer review history of their article (what does this mean? ). If published, this will include your full peer review and any attached files.

**Do you want your identity to be public for this peer review?** For information about this choice, including consent withdrawal, please see our Privacy Policy .

Reviewer #1: No

Reviewer #3: No

---

## [Author Response · Author response to Decision Letter 2]

9 May 2025

Reviewer #1 Comment A.1: Introduction Section. Tighten the narrative flow by placing a clear, concise statement of the research gap earlier in the introduction — ideally before the research questions and objectives are introduced.

AuthorResponse: Thanks for this observation. We have revised the introduction to improve narrative flow by positioning the research gap after the literature review and immediately before the study aim and objectives (p. 3, paragraph 3). This placement allows the gap to emerge logically from prior work, ensuring that it is both well-supported and clearly framed. The paragraph beginning with “However, despite these contributions, research on psychological consultation apps in Saudi Arabia remains sparse” now summarizes the key limitations in the literature and transitions directly into our study’s contribution.

Reviewer #1 Comment A.2: Introduction Section. Clearly state what is missing in the current literature. For example:

• Have previous studies not focused on mental health apps in the Saudi Arabian context?

• Is there a lack of comparative usability analysis using real user reviews?

• Are cultural and contextual usability factors underexplored in prior research?

Author Response: We have explicitly addressed this concern in paragraph 3 of the revised introduction (p. 3). The revised text clearly identifies that:

• Few studies focus specifically on psychological consultation apps.

• There is limited use of real user feedback to assess usability.

• The cultural and linguistic needs of Saudi users are underexplored.

• There is a lack of comparative analysis that integrates heuristic evaluation with sentiment analysis of actual users. These points are presented together to sharpen the articulation of the research gap.

Reviewer #1 Comment A.3: Introduction Section. Follow the gap with a direct explanation of why this study matters: Emphasize how addressing this gap will contribute new knowledge, practical value, or methodological contribution.

AuthorResponse: Thanks for this comment. We have addressed this in paragraph 4 of the revised introduction (p. 3). The study’s relevance is now emphasized by stating that it evaluates three apps (Labayh, Mind, and Estenarah) using a dual-method approach that integrates heuristic evaluation and sentiment analysis. This contributes methodologically by combining expert and user perspectives, and practically by producing culturally grounded recommendations for app designers and healthcare policymakers in Saudi Arabia.

Reviewer #1 Comment B.1: Discussion Section 1) Structural Organization. The discussion section would benefit from a clearer structural organization. Currently, it merges results, analysis, and recommendations in a way that can obscure the key insights of the study. I recommend dividing this section into clear thematic sub-sections, such as:

• Summary of Key Findings

• Usability Challenges (with potential sub-themes like error prevention, interface design, etc.)

• Positive Aspects

• Recommendations for Improvement

• Implications for Mental Health App Design in Saudi Arabia

Author Response: We appreciate this suggestion. We restructured the entire Discussion section (Section 5) into the exact five sub-sections recommended by the reviewer: 5.1 Summary of Key Findings, 5.2 Usability Challenges, 5.3 Positive Aspects, 5.4 Evidence-Based Recommendations, and 5.5 Implications for Mental Health App Design in Saudi Arabia. This reorganization enhances clarity and ensures a more coherent presentation of the study’s insights.

Reviewer #1 Comment B.2: Discussion Section 2) Integration with Literature. While the discussion references relevant literature, there is often a lack of clear integration between the empirical findings and these theoretical insights. I suggest more explicitly linking findings from the current usability evaluations with the literature cited. Example Revision: Instead of stating:

'Research indicates that effective error messages are crucial for user satisfaction...'. It would be stronger to write: “Our findings indicate that Mind’s ambiguous error messages negatively affected user experience. This aligns with Smith et al. [74], who argue that clearly phrased, contextual error feedback enhances user satisfaction and system transparency.”

Author Response: We agree with the reviewer that stronger integration was needed. In the revised Discussion section (particularly 5.2 Usability Challenges and 5.3 Positive Aspects), we enhanced the alignment between empirical findings and relevant literature. For example, we explicitly link Mind’s ambiguous error messaging to prior findings by Nielsen [74], and we expanded this integration to include a wide range of supporting studies. To strengthen this section further, we have now incorporated references [74–96], covering topics such as cognitive load [77], mobile usability heuristics [76], adaptive UI design [78], engagement aesthetics [75, 87, 88], testing practices [82–84], backend optimization [85, 86], and cultural usability in Saudi Arabia [96]. This enriched literature support ensures that the discussion is analytically grounded and theoretically informed throughout.

Reviewer #1 Comment B.3: Discussion Section 3) Comparative Analysis of the Three Apps. The comparative analysis between the three apps—Labyh, Mind, and Estenarah—could be further developed. While comparative statements exist, they often remain descriptive rather than analytical. Expand on why one app succeeded or failed relative to the others...”

Author Response: Thank you for this insightful comment. We enhanced the comparative depth throughout Sections 5.2 and 5.4 by discussing the design choices and implementation strategies that contributed to differences in usability. For example, we analyzed how Labyh’s cleaner navigation contributed to better task performance, while Estenarah’s simplicity came at the cost of feature limitations. These analyses help explain not only what happened but why usability scores diverged between apps.

Reviewer #1 Comment B.4: Discussion Section 4) Repetition of Themes. There is noticeable repetition of some themes, such as task overload, error prevention, and interface challenges, across multiple sections. Consolidating overlapping discussions could enhance clarity and allow more space for nuanced interpretation.

Author Response: We carefully reviewed and revised the entire Discussion section to eliminate repetition. Related topics were consolidated under broader labels within 5.2 Usability Challenges (e.g., error messaging and terminology are now combined under “Error Prevention and Messaging”). This refinement reduces redundancy and enhances thematic clarity.

Reviewer #1 Comment B.5: Discussion Section 5) Descriptive vs Evaluative Tone. Some parts of the discussion adopt a descriptive tone, which weakens the analytical rigor of the section. Suggestion: Adopt a more evaluative tone to critically reflect on the findings.

Author Response: We revised multiple passages across the Discussion to adopt a more evaluative and critical tone. For instance, we now explicitly state how Labyh’s failure to adapt across devices reveals a significant design flaw despite otherwise strong performance. The revised text presents more analytical reasoning, in line with the reviewer’s example.

Reviewer#1 Comment#C: (Recommendation Section). The recommendations section provides a series of general usability best practices — such as improving error messaging, implementing responsive design, and supporting diverse payment options — which are certainly relevant to the design of psychological consultation apps. However, as it stands, this section lacks a clear, direct link to the empirical findings presented in the earlier parts of the paper. While the study includes valuable data on usability issues derived from user reviews and comparative app analysis, these insights are not explicitly reflected in the recommendations. This results in a disconnect between the evidence gathered and the guidance offered. For instance, suggestions such as “conduct user journey mapping” or “implement contextual adaptation” are useful but appear to be generic UX advice rather than recommendations tailored to the actual usability problems uncovered in Mind, Labyh, and Estenarah. Suggestion for Improvement:

To enhance the value and academic rigor of the recommendations, I strongly encourage the authors to restructure this section to reflect the comparative analysis presented earlier. One effective approach would be to build a matrix that clearly maps:

• Usability problems observed in each app

• Comparative insights (e.g., where one app succeeded and another failed)

• Targeted, app-specific recommendations grounded in these observations

Author Response: We appreciate this suggestion and have removed the standalone recommendations section. Instead, we integrated a detailed, evidence-based and app-specific set of recommendations directly into the Discussion section under Subsection 5.4 – Evidence-Based Recommendations. This subsection:

• Links each recommendation explicitly to the findings of the heuristic evaluation and user sentiment analysis.

• Identifies which app(s) the issue pertains to.

• Follows a structured format resembling the suggested matrix (Issue → App → Recommendation).

Reviewer#1 Comment#D: (Conclusion Section). While the conclusion effectively summarizes the key findings, it tends to be overly descriptive and lacks synthesis. Rather than reiterating the results, the conclusion should offer a deeper interpretation of their broader significance — essentially answering the “so what?” of the study. Additionally, the paragraph addresses multiple themes (usability findings, future directions, methodological limitations, sentiment analysis critiques, and reviewer demographics) within a single block, which dilutes clarity and impact. Consider separating the conclusion into clearer thematic segments. Start with a concise summary of the main findings and their implications, followed by a discussion of methodological limitations, and conclude with forward-looking insights or practical recommendations. The unique contribution of the study — such as the integration of heuristic evaluation and sentiment analysis within a culturally specific context (Saudi Arabia) — should be highlighted more explicitly.

Author Response: We appreciate the reviewer’s insightful feedback. In response, we revised the conclusion paragraph to move beyond description and offer a more analytical synthesis of the findings. The revised paragraph now emphasizes the broader significance of the study, clearly articulates its unique methodological contribution, namely (the integration of heuristic evaluation and sentiment analysis in a culturally specific context), and includes a concise reflection on limitations and directions for future work. While we maintained a unified paragraph format for flow, we ensured the paragraph addresses each theme explicitly.

Reviewer #3 Comment#1: Are you sure the sentiment analysis falls under the category of quantitative approach? The categorization to into several sub-themes such as general dissatisfaction, booking and payment issues, access issues, functionality errors, interface concerns, app performance, and feature limitations (for negative reviews) and general appreciation, praise for the app idea, and recommendations for improvement (for positive reviews) based on the subjective reviews strongly indicates qualitative analysis. How are the percentages of each category as shown in Fig 2 and 3 obtained?

Author Response: Thank you for this valuable observation. We acknowledge that the sub-themes identified are interpretable in a qualitative sense; however, we clarify that the sentiment analysis methodology applied in this study is quantitative in nature. Specifically, we used CountVectorizer to convert the preprocessed review texts into a numerical document-term matrix. This matrix was then processed using Latent Dirichlet Allocation (LDA), a probabilistic topic modeling technique that generated topic distributions across the corpus. Each review was automatically assigned to its most probable topic based on posterior probabilities generated by the LDA model. The percentages shown in Figures 2 and 3 reflect the proportional distribution of reviews assigned to each topic. This explanation has now been added to the end of Section 3.2 to clarify the quantitative nature of the analysis and the method used to calculate topic-wise distributions.

---

## [Decision Letter · Decision Letter 2]

PONE-D-24-40629R2Psychological consultation apps in Saudi Arabia: A study for experts’ evaluation and users’ points of viewPLOS ONE

Dear Dr. Albesher,

Thank you for submitting your manuscript to PLOS ONE. After careful consideration, we feel that it has merit but does not fully meet PLOS ONE’s publication criteria as it currently stands. Therefore, we invite you to submit a revised version of the manuscript that addresses the points raised during the review process.

Following further review, the Academic Editor is satisfied with your revisions; please see their comments below. However, before we can proceed with publication, we kindly ask that you thoroughly copyedit your manuscript for language usage, spelling, and grammar. If you do not know anyone who can help you do this, you may wish to consider employing a professional scientific editing service.  

The American Journal Experts (AJE) (https://www.aje.com/) is one such service that has extensive experience helping authors meet PLOS guidelines and can provide language editing, translation, manuscript formatting, and figure formatting to ensure your manuscript meets our submission guidelines. Please note that having the manuscript copyedited by AJE or any other editing services does not guarantee acceptance for publication. 

- A clean copy of the edited manuscript (uploaded as the new *manuscript* file).

We thank you for your attention to this request.

We look forward to receiving your revised manuscript.

Kind regards,

Hugh Cowley

Senior Editor

PLOS ONE

on behalf of

Najmul Hasan, PhD

Academic Editor

PLOS ONE

Journal Requirements:

Additional Editor Comments:

Following a comprehensive review, your paper has been acknowledged for its substantial contribution to the field of health sciences. Your research offers valuable insights into the usability of psychological consultation apps, and we are confident that it will have a significant impact for designers of psychological consultation apps, as well as practical applications aimed at improving the user experience of psychological consultation apps in Saudi Arabia

Reviewers' comments:

Reviewer's Responses to Questions

**Comments to the Author**

1. If the authors have adequately addressed your comments raised in a previous round of review and you feel that this manuscript is now acceptable for publication, you may indicate that here to bypass the “Comments to the Author” section, enter your conflict of interest statement in the “Confidential to Editor” section, and submit your "Accept" recommendation.

Reviewer #1: All comments have been addressed

2. Is the manuscript technically sound, and do the data support the conclusions?

Reviewer #1: Partly

3. Has the statistical analysis been performed appropriately and rigorously? 

Reviewer #1: Yes

4. Have the authors made all data underlying the findings in their manuscript fully available?

Reviewer #1: Yes

5. Is the manuscript presented in an intelligible fashion and written in standard English?

Reviewer #1: Yes

6. Review Comments to the Author

Reviewer #1: While the structure of the revisions could be improved, the authors have addressed all reviewer comments satisfactorily, and the responses are acceptable.

7. PLOS authors have the option to publish the peer review history of their article (what does this mean? ). If published, this will include your full peer review and any attached files.

**Do you want your identity to be public for this peer review?** For information about this choice, including consent withdrawal, please see our Privacy Policy .

Reviewer #1: No

---

## [Author Response · Author response to Decision Letter 3]

26 Jun 2025

Reviewer #1 Comment 1: While the structure of the revisions could be improved, the authors have addressed all reviewer comments satisfactorily, and the responses are acceptable.

Author Response: We thank the reviewer for acknowledging that all comments have been satisfactorily addressed. Regarding the observation about the “structure of the revisions,” we believe this may refer to the consistency and clarity of the revised text following the incorporation of all reviewer suggestions. To strengthen the final presentation, we engaged a professional proofreading service (PaperTrue), which made over 780 language and structure-related improvements to the manuscript (380 insertions, 384 deletions, and 18 comments) across a 6,498-word document. These changes addressed grammar, cohesion, sentence structure, and consistency, while ensuring that the content remained faithful to the revised structure. We hope these enhancements improve the overall clarity and readability of the manuscript and address any remaining structural concerns.

Editor Comment: Following a comprehensive review, your paper has been acknowledged for its substantial contribution to the field of health sciences. Your research offers valuable insights into the usability of psychological consultation apps, and we are confident that it will have a significant impact for designers of psychological consultation apps, as well as practical applications aimed at improving the user experience of psychological consultation apps in Saudi Arabia

Author Response: We sincerely thank you for your kind feedback and for acknowledging the contribution of our work. We are grateful for the opportunity to share our research through your journal and are encouraged by your recognition of its value to the field of health sciences. We hope that our findings will indeed support both researchers and practitioners seeking to improve the usability of psychological consultation apps, particularly within the Saudi Arabian context. We appreciate the editorial guidance and constructive reviewer feedback that helped us refine and strengthen the manuscript.

---

## [Editor Report · Decision Letter 3]

Psychological consultation apps in Saudi Arabia: A study for experts’ evaluation and users’ points of view

PONE-D-24-40629R3

Dear Dr. Albesher

We’re pleased to inform you that your manuscript has been judged scientifically suitable for publication and will be formally accepted for publication once it meets all outstanding technical requirements.

Kind regards,

Najmul Hasan, PhD

Academic Editor

PLOS ONE
---

## [Editor Report · Acceptance letter]

PONE-D-24-40629R3

PLOS ONE

Dear Dr. Albesher,

I'm pleased to inform you that your manuscript has been deemed suitable for publication in PLOS ONE. Congratulations! Your manuscript is now being handed over to our production team.

Kind regards,

on behalf of

Dr. Najmul Hasan

Academic Editor

PLOS ONE